# Studies on Improving the Efficiency of Somatic Embryogenesis in Grapevine (*Vitis vinifera* L.) and Optimising Ethyl Methanesulfonate Treatment for Mutation Induction

**DOI:** 10.3390/plants12244126

**Published:** 2023-12-11

**Authors:** Ranjith Pathirana, Francesco Carimi

**Affiliations:** 1The New Zealand Institute for Plant and Food Research Limited, Batchelar Road, Palmerston North 4472, New Zealand; 2Istituto di Bioscienze e BioRisorse (IBBR), Consiglio Nazionale delle Ricerche, Via Ugo la Malfa, 153, 90146 Palermo, Italy; francesco.carimi@ibbr.cnr.it

**Keywords:** auxin, crop improvement, cytokinin, floral tissue, explant, plant growth regulators, tissue culture

## Abstract

Somatic embryogenesis (SE) has many applications in grapevine biotechnology including micropropagation, eradicating viral infections from infected cultivars, mass production of hypocotyl explants for micrografting, as a continuous source for haploid and doubled haploid plants, and for germplasm conservation. It is so far the only pathway for the genetic modification of grapevines through transformation. The single-cell origin of somatic embryos makes them an ideal explant for mutation breeding as the resulting mutants will be chimera-free. In the present research, two combinations of plant growth regulators and different explants from flower buds at two stages of maturity were tested in regard to the efficiency of callusing and embryo formation from the callus produced in three white grape cultivars. Also, the treatment of somatic embryos with the chemical mutagen ethyl methanesulfonate (EMS) was optimised. Medium 2339 supplemented with β-naphthoxyacetic acid (5 μM) and 6-benzylaminopurine (BAP—9.0 μM) produced significantly more calluses than medium 2337 supplemented with 2,4-dichlorophenoxyacetic acid (4.5 µM) and BAP (8.9 µM) in all explants. The calluses produced on medium 2337 were harder and more granular and produced more SEs. Although the stage of the maturity of floral bud did not have a significant effect on the callusing of the explants, calluses produced from immature floral bud explants in the premeiotic stage produced significantly more SEs than those from more mature floral buds. Overall, immature ovaries and cut floral buds exposing the cut ends of filaments, style, etc., tested for the first time in grapevine SE, produced the highest percentage of embryogenic calluses. It is much more efficient to cut the floral bud and culture than previously reported explants such as anthers, ovaries, stigmas and styles during the short flowering period when the immature flower buds are available. When the somatic embryos of the three cultivars were incubated for one hour with 0.1% EMS, their germination was reduced by 50%; an ideal treatment considered to obtain a high frequency of mutations for screening. Our research findings will facilitate more efficient SE induction in grapevines and inducing mutations for improving individual traits without altering the genetic background of the cultivar.

## 1. Introduction

Somatic embryogenesis (SE) is a powerful biotechnological tool that enables the genetic manipulation of clonally propagated species, including grapevines. So far, SE is the only approach that enables stable genetic transformation of grapevines [1,2,3,4,5]. Furthermore, somatic embryogenesis has been shown to eliminate several important viruses [6,7,8] affecting grapevines including viroid infections [9], and could therefore supplement established virus elimination procedures such as meristem culture, cryotherapy and heat therapy. 

Histological studies have shown that somatic embryos (SEs) can be derived from single cells [10,11,12]. The efficient separation of cell layers in chimeric cultivars such as Pinot Meunier [13], Chardonnay 96 [14], Nebbiolo [15] etc. during SE-based regeneration, as well as the development of non-chimeric transgenic grapevine plants during *Agrobacterium* mediated transformation using embryogenic cultures [16,17] allows SE to be used to understand the genetic makeup of ancient cultivars. On the other hand, chimerism is a frequent problem in the *Agrobacterium* mediated transformation of grapevine embryogenic calluses as demonstrated for example by: Ren et al. [18] and Nakajima et al. [19].

SEs have also been suggested as a potential source of hypocotyl explants for grapevine micrografting [20] and for germplasm conservation, [21,22,23,24] as true seeds do not represent the genotype of the original clonal material. They can be stored under low temperature for medium-term storage [24,25] or cryopreserved for long-term storage [26,27]. In coconut for example, fast growing embryogenic calluses produced through anther culture at the late uninucleate stage of microspore development have been proposed as a source for the long-term supply of haploid and doubled haploid plants [28] and this could be extended to grapevines as anthers at the right stage of development are available only during a short period of the year. SEs are also an excellent source for mutation induction [29,30,31]. The single-cell origin of SEs often reported in cell cultures makes them an attractive option for use in induced mutagenesis overcoming chimeric situations which complicate selection following mutagenesis in conventional mutation breeding programmes [29,32]. SEs can either be germinated directly or made to produce secondary SEs, which arise from single cells [33], making this system even more attractive. The possibility of subjecting large cell populations to mutagens under in vitro conditions is another advantage of using such a system for mutagenesis. Induced mutations have been used in the past to improve disease resistance [34,35], stress tolerance [36], maturity date and plant architecture [37,38] and fruit colour, shape and quality [39,40] in fruit crops. Hence, there is a great potential for improving the popular cultivars of grapevine without changing the flavour and other sought after wine characteristics of this relatively traditional crop with very conservative consumers. 

In the recent past, several explants and media formulations have been used with varying success rates to induce grapevine SEs in vitro. Zlenko et al. [41,42,43] induced SEs in liquid culture media in calluses derived from the petiole explants of tissue cultured grape plants. A combination of β-naphthoxyacetic acid (NOA) and thidiazuron enabled Harst [44] to produce SEs from the leaf explants of three genotypes. However, anther explants produced SEs in many more cultivars than leaf explants in that system [45]. Since these early studies, the most frequently used explants for successful SE in grapevines have been derived from floral tissue. These include ovules [46,47], anthers [7,48,49,50,51,52], pistils [6,21], stigmas and styles [33,53], ovaries [54,55,56,57] and the whole flower [55]. It is clear that the efficiency of SE is dependent on the explant used and the maturity of the explant [45,50,55,58], the media constituents [3,49,51,59,60] and the genotype [20,49,55,59]. 

In the current research, we tested the efficiency of several flower-derived explants and two combinations of plant growth regulators in three popular white grape cultivars rarely used in previous studies for embryogenic potential. We included cut floral buds as explants for the first time in grapevine SE studies. We also optimised suitable doses of ethyl methanesulfonate (EMS) for mutation induction in embryogenic cultures and regenerated putative mutants for screening. 

## 2. Materials and Methods

### 2.1. Treatment of Plant Materials

Inflorescences were harvested before flower opening (Figure 1A,B) from three popular cultivars, Chardonnay, Sauvignon blanc and Riesling, maintained in the collection growing in the greenhouse complex at the New Zealand Institute for Plant & Food Research, Palmerston North, New Zealand (Latitude: 40.3545° S, Longitude: 175.6097° E). Harvested inflorescences were immediately transferred to the laboratory and the upper part of the inflorescence was removed. The lower part was sterilized (Figure 1C) with 70% ethyl alcohol for 45 s followed by a solution of 1% sodium hypochlorite containing 0.2% Tween 20^®^ (Sigma-Aldrich, St. Louis, MO, USA) for 20 min on a gyratory shaker (50 rpm). Floral buds were then washed three times in sterile water, transferred to a fresh, sterile container and incubated at 4 °C for five days [50,55] before aseptic culture (Figure 1D). 

### 2.2. Media, Explants Tested and Experimental Procedure

Explants were dissected under sterile conditions and incubated on solid media (Figure 1E). All the media were based on Nitsch & Nitsch [61] major and minor elements and B5 vitamins [62]. For the initiation of embryogenic cultures, two combinations of plant growth regulators that have so far proven successful effective were tested in the experiments: 2,4-dichlorophenoxyacetic acid (2,4-D—4.5 µM) and 6-benzylaminopurine (BAP—8.9 µM) [55,63] (medium 2337), and β-naphthoxyacetic acid (NOA—5.0 μM) and BAP (9.0 μM [33,53] (medium 2339). Both media were supplemented with 6% sucrose and solidified with Gelrite^®^ (Merck, Rahway, NJ, USA) (0.3%).

Three experiments were conducted to test the effectiveness of different plant growth regulator combinations (media 2337 and 2339) and explants in initiating callusing and SEs from the initiated calluses. In the first experiment, three explant types were tested in the cultivars Riesling and Sauvignon blanc (Appendix A): the upper and lower parts (Appendix A) of the flower buds after cutting them horizontally so that the filaments and either the ovary or the style were cut. The explant was placed with the cut surface touching the media. Alongside these two types of explants, stigma and style explants (Appendix A) [33] were also included in this experiment. 

In the second experiment five explant types were tested in all three grapevine cultivars. Three of the explants tested have been reported to be effective in producing embryogenic cultures in *Vitis vinifera*: anther with filament (Appendix A), pistil (Appendix A; Figure 1F) and whole flower bud (Appendix A) [55]. In addition, we used flower buds cut longitudinally (Appendix A) and horizontally and placed them with the cut surface touching the media. In this experiment only the lower part of the horizontally cut flower bud was used considering better callusing results from previous experiment. This exposed the cut surfaces of the ovary or style, filaments and the petals to media (Appendix A).

In the third experiment, young flower buds (<1.5 mm) were separated from more mature buds (>1.5 mm) (Figure 1A,B) and the anther and filament (Appendix A), ovary (Appendix A), pistil (Appendix A), style and stigma explants were cultured separately to understand if there is an effect of the age of the flower bud on the SE potential when different explants are used for inducing embryogenic cultures. The two stages of flower bud correspond to the pre-tetrad or pre-meiotic (<1.5 mm flower bud), and uninucleate (>1.5 mm) stages of microsporogenesis, respectively [64]. According to the classification of Gribaudo et al. [50], the stages of microsporogenesis in the flower buds correspond to the premeiotic (Stage II) and early uninucleate (Stage V) stages, respectively.

### 2.3. Culture Conditions and Handling of Embryogenic Calluses

The cultures were incubated in the dark at 24 °C for a period of 8 weeks after which the callusing explants were counted and transferred to fresh media. Thereafter the cultures were transferred to fresh media every 4 weeks. Embryogenic calluses produced on medium 2337 with globular embryos were transferred to a medium consisting of Nitsch & Nitsch [61] major and minor elements, B5 vitamins [62] supplemented with NOA (10 µM), BAP (1 µM) and indole-3-acetic acid (IAA—20 µM) with 6% sucrose and 0.25% activated charcoal solidified with 0.3% Gelrite^®^ (Merck) [50,63] (medium 3355). After a second transfer onto the same medium in 4 weeks, the mature embryos produced were used in the experiment for optimising chemical mutagen (EMS) dose for mutation induction. Calluses originating on medium 2339 continued to be subcultured monthly, using harder more granular calluses. Once harder callus status was achieved with globular embryos starting to appear, these calluses were subcultured on hormone-free medium (medium 3355 free of plant growth regulators) maintaining embryogenic potential for many months while the globular embryos matured into the cotyledonary stage. For efficient embryo production, harder calluses were used for subculture. 

### 2.4. Optimising Treatment for Chemical Mutagenesis

SEs (Figure 1G) (~2mm–50 per cultivar) generated on medium 2337 from the immature ovaries and cut floral buds (lower side of the bud) of all three cultivars in Experiment 2 were separated from the calluses and treated with 0, 1, and 2% EMS for 1 h in Petri plates in a preliminary experiment. During the treatment, the plates were shaken on a rotary shaker (50 rpm). The embryos were washed three times with sterile distilled water and transferred to MS liquid medium [65] with half-strength major elements supplemented with 3% sucrose for germination (Figure 1H). As the SEs did not survive in 2% EMS treatment, lower concentrations of EMS: 0.00, 0.01, 0.05, 0.10, 0.25, 0.50, 0.75 and 1% were used in the main experiment. 

### 2.5. Data Collection, Statistical Design and Analysis

The experiments were conducted with a minimum of three replicates per treatment. One Petri plate with a minimum of 25 explants was considered as a replicate in the SE induction experiments. Two Petri plates, wrapped together, were used as a replicate in the case of larger explants such as whole flower, as 25 explants could not be accommodated in a single Petri plate. The replicates were maintained in a randomised complete block design within the culture room. In all the experiments, the number of explants producing calluses after eight weeks in culture in the dark, the number of embryogenic calluses that were transferred to fresh media and those producing embryos after two months (two subcultures every four weeks) were recorded. The results were statistically analysed using Genstat (VSN International, Hertfordshire, UK), fitting a binomial generalized linear model with a logit link, and factors for cultivar, explant type and medium, plus their interactions. The least significant differences were calculated from the standard errors of differences of each pair of means on the logit scale, back transformed and then summarised with an average. 

For the mutagenic dose optimization experiment, 25–30 embryos per plate per cultivar in three replicate plates were used using the eight treatments described in the previous section. The percentage of embryos giving a root or secondary SEs which would ultimately result in at least one rooted plantlet was recorded (Figure 1I). The data were analysed using a generalised linear model.

## 3. Results

### 3.1. Effect of Media on Callus Induction and Somatic Embryogenesis

Of the two media tested, medium 2339 produced calluses (both embryogenic and non-embryogenic) in a larger proportion of explants than medium 2337 (Figure 2). Media differences for callus induction were highly significant in all the three experiments (Appendix A). Similarly, the type of explant used also had a highly significant effect on callus induction in all the three experiments (Figure 2, Appendix A). In the experiment with three explants and two cultivars (Experiment 1), the lower parts of the cut flower buds produced significantly more calluses in medium 2339 in Sauvignon blanc, whereas in Riesling both the upper and lower parts of the cut flower buds produced significantly more calluses than stigma and style explants (Figure 2A). The difference in the callus producing ability on the two media was even more obvious in the second experiment with all the explants recording 3–5 times more calluses in medium 2339 than in medium 2337 in all the three cultivars (Figure 2B). The absence of callusing in the anther explants in medium 2337 in all three cultivars in Experiment 2 (Figure 2B) and mature anthers in Chardonnay and mature pistils in Riesling and Sauvignon blanc in Experiment 3 (Figure 2C) are noteworthy. 

The calluses produced on medium 2339 were generally more watery and loose, whereas those produced on medium 2337 were harder, granular and more embryogenic. As a result, the percentage of calluses giving embryos in medium 2337 was greater in many explants across the three experiments (Figure 3). The medium effect is statistically significant in Experiments 2 and 3 but not in Experiment 1 (Appendix A). A notable exception was the style and stigma explants used in Experiment 1 and 3, particularly in Sauvignon blanc, and immature style and stigma explants in Experiment 3, where medium 2339 gave better callus to embryo conversion rates (Figure 3).

### 3.2. Effect of the Explant, the Age of the Explant and the Cultivar on Callus Induction and Somatic Embryogenesis

The explant effect was highly significant for the number of explants callusing in all three experiments (Appendix A). The type of explant also had a significant effect on the number of embryogenic calluses produced out of the total cultured in Experiments 2 and 3. The effect of the explant and the effect of the maturity of the explant on the proportion of callused explants producing embryos were highly significant in Experiment 3 (Appendix A). The cut flowers produced significantly more calluses than the style and stigma explants in medium 2339 in Experiment 1 (Figure 2A). Similarly, in Experiment 2 also the cut flowers gave higher callusing percentages compared to pistils and the whole flowers in medium 2339, whereas anther explants recorded a similar number of calluses to cut flowers (both transversely and longitudinally cut) in this medium (Figure 2B). Calluses from cut flowers produced relatively more embryos than explants giving smaller calluses such as anthers (Figure 4). The cultivar effects were significant only in Experiment 1 for the proportion of callused explants out of the total cultured (Appendix A), and also at the <0.05 level of probability.

Explant age was studied as a factor in Experiment 3. Although there was no explant age effect on callusing percentage, there were highly significant effects of the age of the explants in the production of embryogenic calluses, both as a percentage of cultured and callused explants (Appendix A). This is due to the high conversion ratio of calluses from immature explants into embryos (Figure 3C). 

### 3.3. Interaction Effects of Media, Genotype and Explant Maturity

Although the majority of interactions were not significant, the significant interaction effect of explant type and medium on callusing percentages in all three experiments is noteworthy (Appendix A). We also found highly significant interactions between the maturity of the explant and the media used on the proportion of callusing explants (Experiment 3, Appendix A) and for all the parameters studied in the 3-way interaction of cultivar, explant maturity and medium (Experiment 3, Appendix A). 

When the percentage of cultured explants producing embryos (i.e., embryogenic calluses) is considered, the lower parts of the cut flower buds cultured in medium 2339 recorded the highest number (10%) in both Riesling and Sauvignon blanc (Appendix A). In the Experiment 2, anthers (18%) and lower parts of the cut flower buds (16%) in medium 2339 recorded the highest average response (Appendix A). In both these experiments, all the explants cultured on medium 2337 resulted in less than 5% of embryogenic calluses. In Experiment 3, where we tested the maturity of the explant as well, the highest percentage of embryogenic calluses per cultured explant was recorded with immature ovaries in all the three cultivars, but in medium 2337 for Chardonnay and Sauvignon blanc and in medium 2339 for Riesling (Appendix A). Due to the large number of treatments, cut flower buds were not tested in this experiment. Somatic embryos produced from the three cultivars in Experiment 2 with cut floral buds cultured on medium 2339 are shown in Figure 5.

### 3.4. Effect of Ethyl Methanesulfonate on Plant Regeneration from Somatic Embryos

As 1 and 2% EMS treatment for 1 h resulted in the high mortality of SEs in our first experiment, we reduced the concentration and tested 0.01–1% concentrations. With the increase in concentration, the germination percentage of SEs reduced in all three cultivars (Table 1). Statistical analysis using a generalised linear model showed that there were no significant differences between cultivars in their reaction to the EMS dose. In all three cultivars 0.1% EMS for one hour resulted in a 50% reduction in germination (Table 1).

## 4. Discussion

This study compared the efficiency of two protocols widely used for inducing *V. vinifera* embryogenic cultures using different explants of floral origin harvested at two stages of development. In addition to traditionally used explants such as anthers, ovaries, styles/stigmas and whole flower buds, we tested flower buds cut in half for their embryogenic potential. We hypothesized that the exposure of the cut surface of the flower buds to the media would allow for better callusing than intact flower buds. This is due to the great capacity that plants have, to react to tissue injury, in our case the damage to floral tissues. In plants, mature and differentiated cells have the ability to change their function through dedifferentiation and the subsequent regeneration of damaged tissue [66]. Moreover, it is much easier and efficient to establish large numbers of cultures in a relatively short period with this explant. This is important, as grapevine flower buds of suitable maturity are available only during a short period of 3–4 weeks during the year [67]. In both experiments where we used the cut flower, embryogenic callus cultures were initiated successfully. Medium 2339 containing NOA and BAP produced calluses in a larger proportion of explants but with less embryogenic potential and the reverse was true for medium 2337 with 2,4-D and BAP. Proportionately less calluses were produced on medium 2337 than in medium 2339, but they were harder and granular, thus giving proportionately more embryogenic calluses than in medium 2339.

Another aspect we studied was the effect of the age of the flower bud on callus and embryo production efficiency. This again is important when the flowering period is short, such as in the case of grapevines. The results were interesting in that age does not affect the callusing frequencies of the explants, but the ratio of calluses that convert to embryos does depend on the age of the explant, with calluses derived from younger flower buds giving embryos more often. Hence, the use of younger flower buds is important for any explant. The need to use flower buds in the early stages of development makes the harvesting window for culture establishment even narrower, and hence the more efficient method of using floral bud explants cut in half is a useful finding. We measured only the flower bud length to discriminate between old and young flower buds because Cardoso et al. [64] have shown a high correlation of flower bud length with anther and ovary development (r = 0.83) in cv. Aragonez and Bouamama et al. [48] confirmed this in nine diverse cultivars. The two stages correspond to the premeiotic and uninucleate stages of pollen development as described by Gribaudo et al. [50] and Cardoso et al. [64]. Immature ovaries recorded the highest percentage of embryogenic cultures in Experiment 3 where we did not test cut flower buds. When cut flower buds were also tested in the two previous experiments, the lower parts of the cut flower buds containing cut anther filaments, immature ovaries and petals recorded the highest percentage of embryogenic calluses.

SE in grapevines has been reported by many authors but the results are highly variable. The reasons for these differences appear to be the use of different floral tissues, as they have shown the capacity to undergo SE with variable success rates, as also shown in our results. The differences in the genotypic responses [49,55,59,68,69] and media differences [22,48,69] also add to this observed variable response. On the contrary, the three cultivars that we used did not show significant differences in their ability to induce calluses or SEs. However, Perl et al. [70], Faure et al. [71], Franks et al. [13], Leal et al. [72] and Cutanda et al. [73] observed variable responses of different cultivars to treatments with different combinations of auxins and cytokinins using 0.5–1 mm long anthers (10–14 days before anthesis) using a combination of 2,4-D and BAP to initiate calluses. Perrin et al. [68,69] used clusters of anthers (five anthers from each flower) to improve callogenesis in optimised media. They demonstrated the importance of combinations of macro and micro elements in the media for optimising embryogenesis in 17 cultivars. Similar to our results with freshly cut floral buds, they demonstrated that anthers with cut filaments produce much more embryogenic calluses compared to anthers detached from the calyx without cutting [68].

Cut flower buds have not been tested so far in grapevine SE. We thought of introducing this treatment, as it was reported that it is the cut surfaces of anthers or ovaries that produce initial calluses that become embryogenic later. The detached ends of filaments (in contrast to cutting) produced much less calluses [68]. Exposing the cut surfaces to growth substances, our method of culture is much faster and efficient than extracting ovaries or anthers under a binocular microscope.

When using medium 2337 combining BAP and 2,4-D, embryos could be produced in 3 months and globular embryos could be matured on medium 3355. For the embryos produced on medium 2339, we used MS medium with half strength major elements for germination. Embryos produced on both media were used in the EMS treatment. EMS is the most widely used chemical mutagen in mutation breeding, considering its high effectiveness and ease of handling (ease of detoxification through hydrolysis for disposal) compared to other chemical mutagens of the same group such as nitroso compounds [29,32,74] or physical mutagens that result in chromosomal aberrations [29,30,75,76]. Therefore, we optimised the dosage of EMS for mutation induction in the three valuable cultivars that we used in our somatic embryo induction studies. It is considered that a 50% lethal dose is optimal for inducing a high rate of mutations covering the genome of the species of interest [29,30,77,78]. Our testing with the three cultivars revealed that treating with 0.1% EMS for one hour reduces the germination of SEs by 50% in all three of the cultivars studied, thus giving opportunity to improve these clones with the desired traits without adversely affecting wine quality, as EMS induces point mutations. There were no significant differences among the three cultivars for sensitivity to EMS at this concentration, just as they behaved similarly in the embryogenic callus induction experiments.

## 5. Conclusions

In the present research we tested two widely used combinations of plant growth regulators for the efficiency of SE induction in three popular white grapevine cultivars using explants derived from flower buds at two stages of maturity. When the medium was supplemented with NAA (5 μM) and BAP (9.0 μM), significantly more calluses were produced compared to the medium supplemented with 2,4-D (4.5 µM) and BAP (8.9 µM) in all explants and cultivars. However, the latter combination produced harder calluses with a higher embryogenic potential resulting in significantly more embryogenic calluses than the calluses produced using the former combination. The stage of the maturity of the floral buds used to obtain the explants did not have a significant effect on callusing, but calluses produced from immature floral bud explants in the premeiotic stage produced significantly more SEs than those from more mature floral buds. Among the many explants tested, immature ovaries and the cut flower buds that were tested for the first time in grapevines, produced the highest percentage of embryogenic calluses. The establishment of cultures using cut floral buds in the pre-meiotic stage is much more efficient than other explants during the short period of the occurrence of this stage during the grapevine flowering period.

We also optimised the treatment of somatic embryos with the chemical mutagen EMS, with a 50% reduction in SE germination after treatment with 0.1% EMS for one hour, an ideal treatment considered to produce a high frequency of point mutations for screening. Our findings have the potential to improve the efficiency of SE production in grapevines which has many practical applications. The optimised mutagenic treatment will facilitate the improvement of individual traits without altering the genetic background of traditional grapevine cultivars.

## Figures and Tables

**Figure 1 plants-12-04126-f001:**
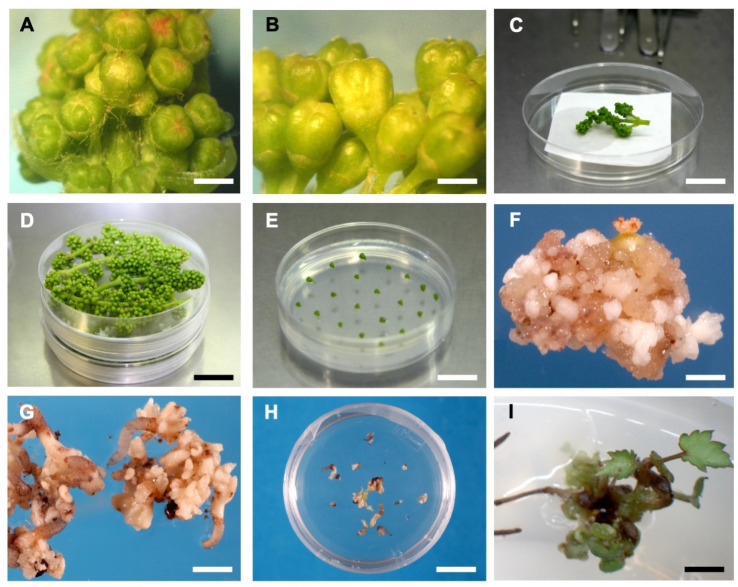
Different stages of somatic embryo (SE) production and mutagenic treatment of SEs. (**A**) Young flower buds (<1.5 mm) (Bar = 1 mm); (**B**) Mature flower buds (>1.5 mm) (Bar = 2 mm); (**C**) Inflorescence sterilized in a laminar flow hood (Bar = 2 cm); (**D**) Sterilized flowers were transferred to a fresh, sterile container and incubated at 4 °C (Bar = 2 cm); (**E**) Whole flower buds incubated on medium 2339 (Bar = 1.5 cm); (**F**) Embryogenic callus generated from immature pistil explant (Bar = 1 mm); (**G**) SEs generated from cut flower (Bar = 1 mm); (**H**) SEs separated from callus and treated with 0.25% ethyl methanesulfonate (EMS) (Bar = 1.5 cm); (**I**) Plantlet derived from a somatic embryo surviving after treatment with 0.25% EMS (Bar = 3 mm).

**Figure 2 plants-12-04126-f002:**
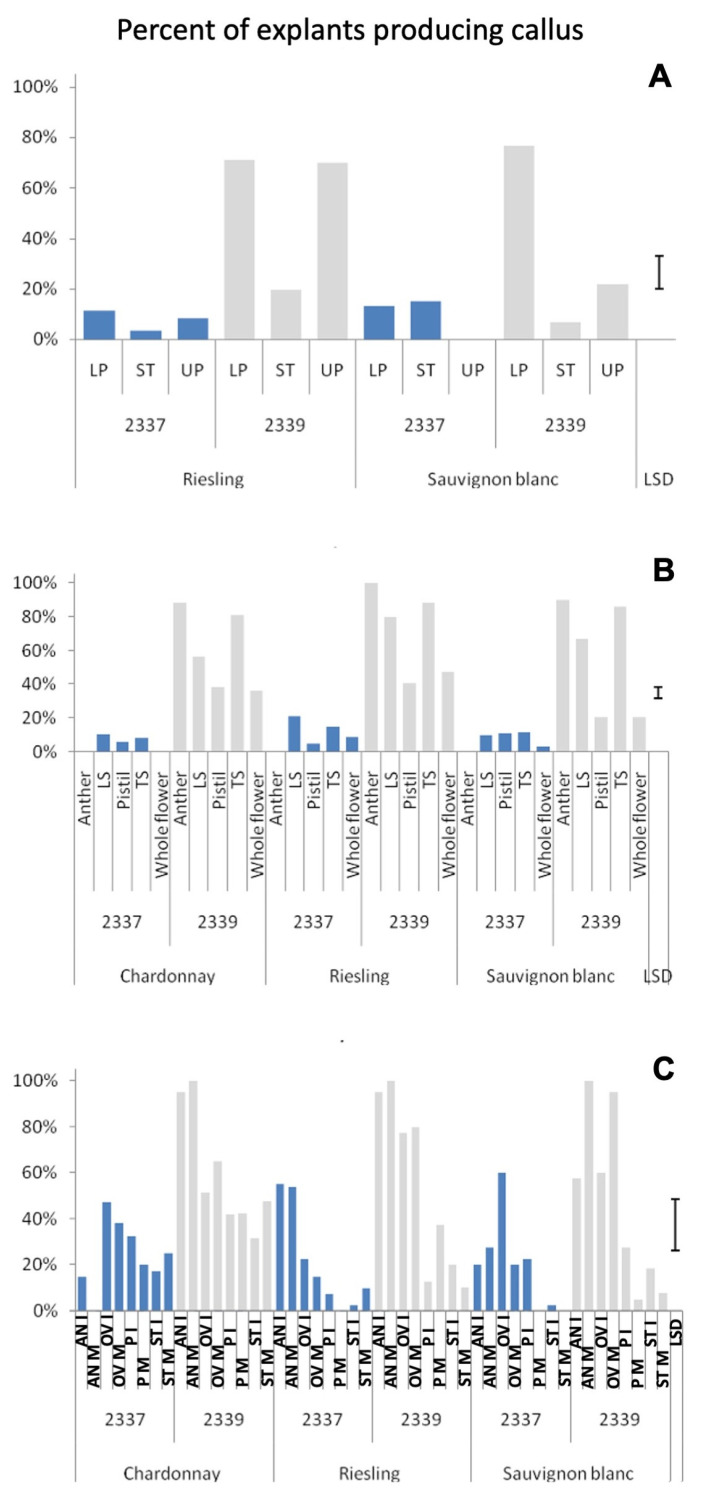
Percent of cultured *Vitis vinifera* cultivar explants producing calluses in two media. (**A**)—Experiment 1; (**B**)—Experiment 2; (**C**)—Experiment 3. LS—Flower bud cut longitudinally, TS—Lower part of flower bud cut horizontally, AN—Anther, OV—ovary, ST—style and stigma, LP—Lower part, UP—upper part, P—Pistil, I—immature (explant taken from a young flower bud < 1.5 mm) and M—mature (explant taken from a larger flower bud > 1.5 mm). LSD—Least Significant Difference.

**Figure 3 plants-12-04126-f003:**
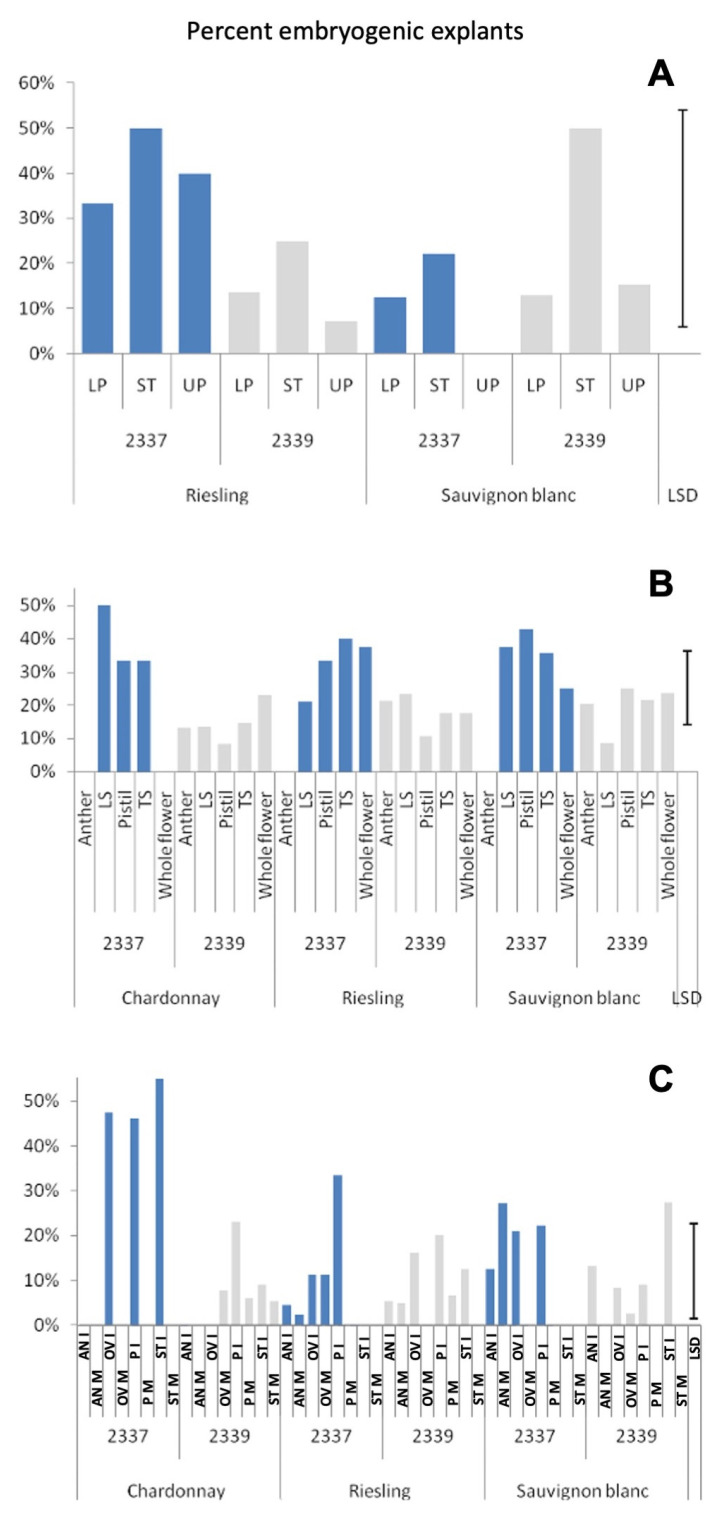
Percent of callused explants (total number of explants with calluses) producing somatic embryos in two media in three white grape cultivars. (**A**)—Experiment 1; (**B**)—Experiment 2; (**C**)—Experiment 3. LS—Flower bud cut longitudinally, TS—Lower part of flower bud cut horizontally, AN—Anther, OV—ovary, ST—style and stigma, LP—Lower part of cut flower, UP—upper part of cut flower, P—Pistil, I—Explant from immature flower bud (flower bud < 1.5 mm) and M—Explant from mature flower bud (flower bud > 1.5 mm). LSD—Least Significant Difference.

**Figure 4 plants-12-04126-f004:**
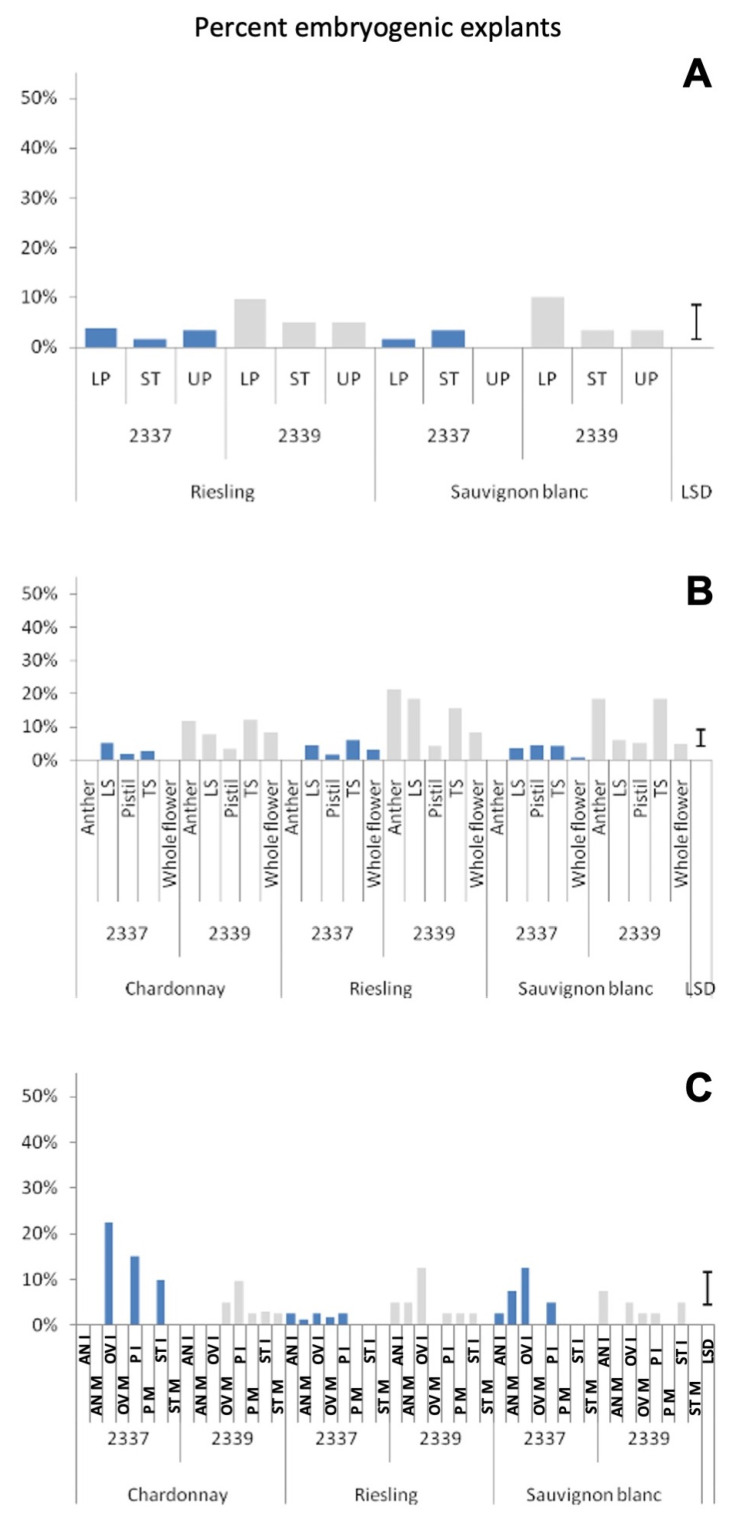
Percent of cultured explants producing somatic embryos in two media in three white grape cultivars. (**A**)—Experiment 1; (**B**)—Experiment 2; (**C**)—Experiment 3. LS—Flower bud cut longitudinally, TS—Lower part of flower bud cut horizontally, AN—Anther, OV—ovary, ST—Style and stigma, LP—Lower part of cut flower, UP—upper part of cut flower, P—Pistil, I—Explant from immature flower (flower bud < 1.5 mm) and M—Explant from mature flower (flower bud > 1.5 mm). LSD—Least Significant Difference.

**Figure 5 plants-12-04126-f005:**
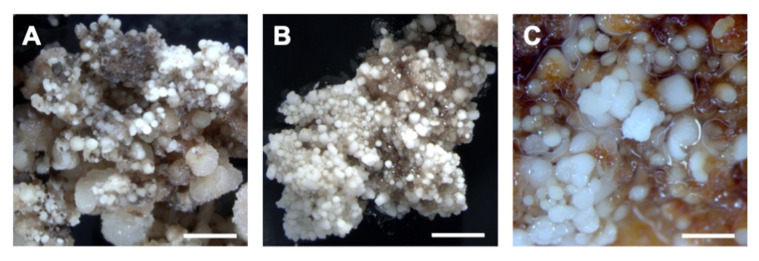
Somatic embryos of: (**A**) Sauvignon blanc (bar = 2 mm); (**B**) Chardonnay (bar = 3 mm) and (**C**) Riesling (bar = 1 mm) cultivars induced on media 2339 using cut flowers as the explant.

**Table 1 plants-12-04126-t001:** Effect of ethyl methanesulfonate (EMS) concentration on the germination of somatic embryos of three white grape cultivars: 25–30 somatic embryos were treated per replicate for one hour (*n* = 3). Data were analysed using a generalised linear model.

EMS (%)	Chardonnay		Riesling		Sauvignon Blanc	
	Prediction (%)	SE (%)	Prediction (%)	SE (%)	Prediction (%)	SE (%)
**0**	80	6	69	5	77	4
**0.01**	80	5	65	5	70	5
**0.05**	**59**	6	71	5	68	5
**0.1**	**55**	6	**59**	5	**54**	5
**0.25**	**49**	6	37	5	**47**	6
**0.5**	30	5	28	5	28	5
**0.75**	21	5	13	4	11	3
**1**	16	4	5	2	7	3

Note: SE = Standard error; Prediction values in **bold** are not significantly different from 50% (i.e., 50% ± 2% SE).

## Data Availability

Data are contained within the article and Appendix A.

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
