# Peer review of "Studies on Improving the Efficiency of Somatic Embryogenesis in Grapevine (Vitis vinifera L.) and Optimising Ethyl Methanesulfonate Treatment for Mutation Induction"

_plants, 2023, doi:10.3390/plants12244126_

Round 1
Reviewer 1 Report
Comments and Suggestions for Authors
The goal of this work was to describe the effects of two media, several flower explants and the age of flowers on the calluses and embryos production from three grapevine cultivars. The cutting of flower buds was also included. Moreover the effect of ethyl methanesulfonate treatment on embryos survival has been tested.
Although the work is interesting, the major changes and explanations are required. The suggestions and comments have been added directly into the attached pdf document.

Author Response
Authors (A) response to Reviewer 1 (R)
The comments in the text by Reviewer 1 have been copied and our response follows.
R
The somatic embryo derived from a single cell is not usually in grapevine. Explain in which cultivar, conditions ecc. this happens
A
We agree that there are no straight forward papers demonstrating this in grapevine. Instead, we included three papers demonstrating this in other species and changes the sentence.
R
Chimerism is a rather frequent problem in the Agrobacterium mediated transformation of grapevine embryogenic callus as demonstrated for example by: Ren, C., Liu, X., Zhang, Z. et al. CRISPR/Cas9-mediated efficient targeted mutagenesis in Chardonnay (Vitis vinifera L.). Sci Rep 6, 32289 (2016). https://doi.org/10.1038/srep32289;
Nakajima I, Ban Y, Azuma A, Onoue N, Moriguchi T, Yamamoto T, et al. (2017) CRISPR/Cas9-mediated targeted mutagenesis in grape. PLoS ONE 12(5): e0177966. https://doi.org/10.1371/journal.pone.0177966
A
Thank you. We have added sentence: “On the other hand, chimerism is a frequent problem in the Agrobacterium mediated transformation of grapevine embryogenic callus as demonstrated for example by: Ren et al [18] and Nakajima et al.[19].”
R
Also in grapevine? In which cultivars?
A
This is an example from coconut and we have edited the sentence to be more specific and its applicability to grapevine.
R
How long were the floral buds incubated at 4 °C?
A
We have added the duration – five days.
R
Is it germinating embryo or plantlet?
A
We have changed the caption to suit the photograph of the germinated embryo
R
It would be appropriate to show photos showing the flower parts and explants tested.
A
We have added Supplementary Figure 1 illustrating all the used explants and referred to those in text. Thank you for the suggestion to improve the paper.
R
What parts of the flower did it contain?
A
We changed/added: “In this experiment only the lower part of the horizontally cut flower bud was used considering better callusing results from previous experiment. This exposed the cut surfaces of the ovary or style, filaments and the petals to media (Supplementary Figure S1B).”
R
How were the stages of microsporogenesis verified?
A
We have referred to Gribaudo et al (2004). They have tested two V. vinifera cultivars and the length of floral but perfectly corresponds to the stages of microsporogenesis. One variety they used is Chardonnay also used in our experiment, and the other two varieties we used did not differ in developmental stages from Chardonnay.
R
is a liquid medium without sucrose?
A
Thank you for pointing out the lapse. We have included 6% sucrose in the sentence and the solidifying agent.
R
What was the composition of this medium?
A
It is same as the previously mentioned medium in the same paragraph. We have further clarified it in the revised version of the manuscript.
R
You mean by type of explants? You should specify better
A
By a treatment it is meant that each of the treatments tested. For example in Expt 1 – cultivar, explant and medium. We believe it is understood. To make it clearer, we added that for all the three experiments, we used three replicates per treatment.
R
Describe the post-hoc analysis carried out and whether the data have been normalised
A
We have now included more details of the statistical analysis carried out.
R
Callus originated from which medium and explant type?
A
The origin of SEs for mutagenic treatment added in text.
R
All three?
A
Thank you for this question. At 2% EMS none of the SEs survived and only very few in 1%. In 0% they all survived. So, we added 2% in the sentence.
R
What callus type? Embrygenic, non-embriogenic or total?
A
Both, we mentioned that in the revised sentence in the revised version.
R
What can be the explanation for this absence of callusing?
A
This question appears in Results section. We have explained that explants from more mature flower buds and in medium 2337 have less callusing in the discussion.
R
The letters are not very legible
A
We have improved the lettering in the figures. Thank you.
R
Two questions in caption to Fig 2
A
Both clarified
R
This table should be moved into supplementary file. In general, this table is difficult to read, confusing and dispersive. It seems to be such as the output of the program.
In the article the table 1 should be drawn up in which the significant differences between the variables (and their interactions) are clearer and more immediate.
A
We agree that the Table 1 should have been in Supplementary Material. We have now moved it to supplementary material. Thank you.
R
Usually the asterisk indicates significant differences between significant variables or interactions (* when p<0.05; ** when p<0.01 etc.).
The meaning of these notes is unclear.
A
Yes, we agree. Now we have used a, b, and c in place of asterisks. Thank you.
R
The letters are not very legible
A
The figure has been made more legible.
R
Percent of embryogenic callus?
A
Caption for figure 3 – The percentage was calculated from total number of callused explants. So we further clarified it in the Caption.
R
Better describe the data in Figures 2, 3 and 4, and in Table 1 which is not easily understood. It is unclear which variable is significant and what is significant.
A
We are referring to the age of explant in Experiment 3. The P values are shown in Table 1 – now Supplementary Table 1.
R
The letters are not very legible
A
We have improved the X-axis in Fig. 4 also. Thanks
R
The definition of LS is missing
A
We have now defined LS as Longitudinal section of flower bud in the caption to Fig. 4. Thank you.
R
It is not clear from where the embryos originate. From embryogenic calluses, directly from the explants or from where?
A
We have described the methodology of callus induction and then the induction of somatic embryos in materials and methods. Also, we describe separately about the number of explants producing callus, and out of these how many produce embryos; what we call embryogenic callus. In this section we give results on the percentage of explants that produce somatic embryos from the those explants with callus. The section starts with “When the percentage of cultured explants producing embryos (i.e., embryogenic callus) is considered,…..” and we hope this clear to the reader.
R
Uniform the cells, align the values to make the table clearer. Write the definition of s.e.
A
We have completely overhauled the table (Table 2, now Table 1), to make it much clearer than before. se has been defined in a footnote
R
Was the number of embryogenic calluses lower in 2339 and higher in 2337, while the number of non-embryogenic calluses was higher in 2339 than in 2337? is unclear
A
We have revised the sentences in this section to be clearer.
R
in which cultivar?
A
We have named the cultivar in Cardoso et al. (2010) used and some more detail about the results from other researchers for use of length of floral bud to determine the maturity status. detail.
R
How can this different behavior of the explants in these experiments be explained?
A
This comment refers to the sentence “Interestingly, immature ovary recorded the highest percentage of embryogenic cultures in the Experiment 3 where we did not test the cut flower bud. When the cut floral buds were also tested in the two previous experiments, lower part of the cut flower bud along with anthers produced the highest percentage of embryogenic callus.” We meant anthers contained inside the cut flower. We have changed the sentence to avoid misunderstanding.
R
complete the reference
A
The details have been added

Reviewer 2 Report
Comments and Suggestions for Authors
The paper describes the possibility of increasing the efficiency of somatic embryogenesis in in vitro grape cultures. The possibility of EMS solution mutation will certainly be of interest to breeders and biotechnologists working with this species. Perhaps it will be a useful protocol. However, I have a few questions and suggestions.
1. in summary, the notation of the growth regulators used could be standardised (either full name or abbreviation)
2. in M&M 2.2 whether the addition of Fe- EDTA was compatible with B5 or MS medium, because take is a citation [59]?
3.in 2.2 M&M were preliminary experiments carried out?
4.could you please explain the name of this medium? (2337...)
5.why only 2 out of 3 cultivars were selected for experiment 1
6.In the study, the dose of EMS mutagen was determined. Were there visible phenotypic changes after the mutagen application? Further research could be carried out to verify this.
Author Response
Thank you for the positive assessment of our manuscript. We have taken all the comments into consideration and revised the manuscript. Our responses to the comments are below:
R - Reviewer A - Authors
R -The paper describes the possibility of increasing the efficiency of somatic embryogenesis in in vitro grape cultures. The possibility of EMS solution mutation will certainly be of interest to breeders and biotechnologists working with this species. Perhaps it will be a useful protocol. However, I have a few questions and suggestions.
- in summary, the notation of the growth regulators used could be standardised (either full name or abbreviation)
A - We have given the full name of plant growth regulators first and abbreviated where it is used again.
R- 2. in M&M 2.2 whether the addition of Fe- EDTA was compatible with B5 or MS medium, because take is a citation [59]?
A - We agree that it is not necessary to give a reference to Fe-EDTA. We have removed the reference and Fe-EDTA from text as iron is a major chemical anyway and is present in the already cited media.
R - 3.in 2.2 M&M were preliminary experiments carried out?
A - Yes we conducted a preliminary experiment with 1% and 2% EMS. As there was very low percent of treated somatic embryos germinated in 1% solution and none in 2% solution, we didn't give the data.
R - 4.could you please explain the name of this medium? (2337...)
A- The medium was formulated by us and the number given is its number in Plant & Food Research tissue culture media database. We refer to it by the number and hope it is OK. In previous publications also we used the data base number, for example here: DOI 10.1007/s11240-005-9057-z
R - 5.why only 2 out of 3 cultivars were selected for experiment 1
A - It was the first experiment we planned and when conducting it we realised that we can increase the number of varieties when we test other explants.
R - 6.In the study, the dose of EMS mutagen was determined. Were there visible phenotypic changes after the mutagen application? Further research could be carried out to verify this.
A - We didn't have continued funding to screen the mutants. We exflasked some putative mutants and these were donated to a nursery in Marlborough, New Zealand. Unfortunately, we did not follow it up because of the distance (work conducted in North Island and plants established in South Island) change of priorities in research.
Reviewer 3 Report
Comments and Suggestions for Authors
Dear editor
The present manuscript has a good introduction containing the necessary background, with necessary information looking up in history and current research.
The research tested combinations of plant growth regulators in three popular white grape cultivars which rarely had used in previous studies for embryogenic potential.
Cut floral bud were used as explants for the first time in grapevine SE studies; optimized doses of ethyl methane sulfonate were used for mutation induction in embryogenic cultures.
The methods are clearly described, and the statistical evaluation is OK. There are all the necessary issues in the attached .pdf.
The data is clearly presented.
The figures and the tables must be improved (the format). They are not clear.
There are very representative photos of all the experimentation stages.
I think that authors could involve a paragraph involve a paragraph for in vitro culture conditions and one more for Statistical Analysis.
Conclusions are also necessary.
Regarding the literature, there are recent references, the reference list being OK.
However, I think that the authors could discuss more on two points (see attached .pdf).
The article adds significant data to the current literature, and it could be presented as a printed manuscript in the present SI of Plants after major revision.
*Manuscript lines are not numbered, hence I edited my notes in the attached .pdf

Author Response
R- Reviewer comment A - Authors' response
R - The present manuscript has a good introduction containing the necessary background, with necessary information looking up in history and current research.
A - Thank you
R - The research tested combinations of plant growth regulators in three popular white grape cultivars which rarely had used in previous studies for embryogenic potential.
Cut floral bud were used as explants for the first time in grapevine SE studies; optimized doses of ethyl methane sulfonate were used for mutation induction in embryogenic cultures.
A - Yes. Thank you.
R - The methods are clearly described, and the statistical evaluation is OK. There are all the necessary issues in the attached .pdf.
A - We have addressed the issues raised in the PDF. Thank you for the instructions.
R - The data is clearly presented.
The figures and the tables must be improved (the format). They are not clear.
A - Thank you for the positive assessment of data presentation, we have improved the figures and tables as suggested.
R - There are very representative photos of all the experimentation stages.
I think that authors could involve a paragraph involve a paragraph for in vitro culture conditions and one more for Statistical Analysis.
A - Thank you for the suggestion to improve the Materials and Methods section. We have now incorporated separate sections for culture conditions and statistical analysis of data.
R - Conclusions are also necessary.
Regarding the literature, there are recent references, the reference list being OK.
A - In the Instructions to Authors in Plants website, Conclusion section is optional. As the Reviewer feels the need, we have now added this new section summarising the findings and their potential impact.
R -However, I think that the authors could discuss more on two points (see attached .pdf).
The article adds significant data to the current literature, and it could be presented as a printed manuscript in the present SI of Plants after major revision.
*Manuscript lines are not numbered, hence I edited my notes in the attached .pdf
A - Thank you for the comments in PDF and we apologise for not including line numbers. We have taken the annotations in the PDF to account in revising the manuscript and changed text accordingly as seen from the track changes. Where the Reviewer has questions in PDF, we answer below:
R - (Materials and Methods Section) 2337 contains 8.9 μΜ ΒΑP.
2339 contains 9.0 μΜ ΒΑP.
I am wondering why does this happen?
Could you explain?
A - These media were based on previously published successful protocols. We used the respective concentrations published in the cited papers.
R - in Results Section suggests transfering this to discussion "Absence of callusing in anther explant in medium 2337 in all three cultivars in Experiment 2 (Figure 2B) and mature anthers in Chardonnay and mature pistil in Riesling and Sauvignon blanc in Experiment 3 (Figure 2C) are noteworthy."
A - We wish to retain this sentence in Results section because we have presented full results in Figure, but writing in text only the noteworthy results - the highest and lowest.
R - In footnote to Table 1 we have this text: *because there were no calluses in some treatments, the Cultivar.Explant.Medium effect has 1 df
and the residual 20 df.
A- Table 1 has 1 df for some interactions and we give the reason as a footnote.
Round 2
Reviewer 1 Report
Comments and Suggestions for Authors
Dear Editor,
The authors have answered my questions and revised the manuscript.
Thank you.
Reviewer 3 Report
Comments and Suggestions for Authors
Dear Editor
The authors revised the manuscript and they answered at my questions.
Thank you.